# Targeted Intervention to Reduce Smoking among People with Severe Mental Illness: Implementation of a Smoking Cessation Intervention in an Inpatient Mental Health Setting

**DOI:** 10.3390/medicina56040204

**Published:** 2020-04-24

**Authors:** Julia M. Lappin, Dennis Thomas, Jackie Curtis, Stephen Blowfield, Mike Gatsi, Gareth Marr, Ryan Courtney

**Affiliations:** 1South Eastern Sydney Local Health District, Sydney, NSW 2031, Australia; Jackie.Curtis@health.nsw.gov.au (J.C.); Stephen.Blowfield@health.nsw.gov.au (S.B.); Mike.Gatsi@health.nsw.gov.au (M.G.); Gareth.Marr@health.nsw.gov.au (G.M.); 2School of Psychiatry, Faculty of Medicine, UNSW, Sydney, NSW 2031, Australia; 3National Drug and Alcohol Research Centre, UNSW, Sydney, NSW 2031, Australia; dennis.thomas@newcastle.edu.au (D.T.); r.courtney@unsw.edu.au (R.C.)

**Keywords:** severe mental illness, smoking, smoking cessation, system change intervention, mental health, inpatient

## Abstract

*Background and Objectives:* Smoking and smoking-related harms are highly prevalent among people with severe mental illness. Targeted smoking cessation programs are much needed in this population. This pilot study aimed to assess the effectiveness of implementing smoking cessation system change interventions within an acute inpatient mental health unit. *Materials and Methods:* Design: Pre-post intervention study. System change interventions for smoking cessation were delivered over a three-month period (05 March 2018–04 June 2018) on an acute inpatient mental health unit. Participants (n = 214) were all individuals receiving care as inpatients during the three-month intervention. Outcomes assessed pre- and post-intervention were: (i) recording of patient smoking status in medical notes, (ii) number of inpatients offered smoking cessation medication, and iii) number of violent incidents reported. *Results:* Recording of smoking status significantly increased from 1.9% to 11.4% (X^2^ = 14.80; *p* ≤ 0.001). The proportion of inpatients offered smoking cessation treatment significantly increased from 11.0% to 26.8% (X^2^ = 16.01; *p* ≤ 0.001). The number of violent incidents decreased by half, which was not statistically significant. *Conclusion:* Evidence-based smoking cessation interventions can be successfully implemented on an inpatient mental health unit. Modest gains were made in routine screening for smoking and in smoking cessation treatment prescription. Future studies should prioritize effective participatory collaboration with staff to optimize effectiveness of interventions and should include additional strategies such as brief intervention training and smoking cessation treatments such as varenicline and buproprion in addition to nicotine replacement therapy (NRT).

## 1. Introduction

Smoking prevalence among psychiatric populations varies, with much higher rates among those with severe mental illnesses, such as schizophrenia and psychosis spectrum disorders and bipolar disorder than among the general population [1,2,3]. Despite a steady decline in smoking rates in the general populations of high-income nations, the prevalence of smoking among people with severe mental illness has not reduced over the past several decades [4]. Indeed, approximately 60% of people who experience psychotic illness smoke [1,3,5]. Smoking prevalence is also raised among populations with other mental health issues, such as serious psychological distress, dementia, and phobias. US data from the 2007 National Health Interview Survey showed that 34.3% of people with phobias or fears smoked, compared with 18.3% among the general population with no mental illness [1]. Moreover, people with mental illness are more likely than the general population to smoke more heavily and suffer from smoking-related illnesses [6]. 

The reasons underlying these very high rates are not well understood, though a number of possible explanations include shared genetic or environmental factors (such as social deprivation) that increase risk for both mental illness and smoking [7]; self-medication with nicotine to lessen symptoms of severe mental illness [8]; and a possible causal link between tobacco smoking and the development of new onset psychotic disorders [9]. A further important consideration in the maintenance of these persistently high rates is that people with severe mental illness are known to have inadequate access to smoking cessation services [10]. 

Smoking-related illnesses including cardiovascular disease and cancers contribute significantly to the excess mortality and morbidity suffered by people experiencing severe mental illness [11,12]. Individuals with severe mental illness have both higher rates of cancers and higher case fatality rates from cancers [12,13]. 

Among people experiencing severe mental illness, tobacco smoking is a modifiable risk factor for poor physical and mental health and thus a key priority for intervention. One opportunity to offer smoking cessation is while a person experiencing severe mental illness is an inpatient on a mental health unit. In this setting, if no-smoking policies are enforced, smokers will quickly experience symptoms of nicotine withdrawal, and nicotine replacement therapy (NRT) can assist to reduce and alleviate nicotine withdrawal symptoms. Previous research indicates that it is feasible to support inpatient smokers to temporarily abstain from smoking or to quit [14]. This may be achieved both by offering NRT on admission to avoid symptoms of withdrawal and by providing education on the benefits of NRT [14]. An additional benefit of effectively applying smoking cessation interventions on inpatient mental health units is a reduction in violent incidents [15], challenging the widely-held concern that smoking may help prevent aggression in inpatient settings. Concerns that there may be increased aggression, for example because cigarettes may no longer be used to assist in informal de-escalation and behavioural management, or because smokers will quickly experience symptoms of nicotine withdrawal including agitation, have not been upheld. Indeed, there is evidence from a growing number of studies conducted in inpatient mental health settings that both physical and verbal violence decrease following the introduction of smoke-free policies [15]. Thus, there is a need to recognize that inpatient settings where smoking is commonplace can often reinforce smoking behaviours and the consequent smoking-related harms, and result in relapse to smoking in persons who have previously quit [16]. Informal ward practice may contribute to this through use of cigarettes and smoking breaks to deescalate acute distress and to reward behaviours such as medication compliance [16]. Staff training has been highlighted as a key factor in the success of smoke-free initiatives [17], while obstacles include both a lack of consistency in applying policies, and a culture of acceptance of smoking behaviours [18].

This report details findings from a pilot implementation study which aimed to assess the effectiveness of introducing smoking cessation interventions following training for all staff in key processes including routine prescription of NRT on an acute inpatient mental health unit in improving the following outcomes:

(1) Number of patients with smoking status recorded in medical notes;

(2) Number of patients offered smoking cessation medication;

(3) Number of violent incidents reported on the unit.

## 2. Materials and Methods

### 2.1. Design

Pre-post intervention study. Sample: Participants were all individuals who were receiving care in an acute inpatient mental health setting in the South Eastern Sydney Local Health District, New South Wales, Australia. System change interventions for smoking cessation were delivered over a three-month period (05 March 2018 to 04 June 2018). The intervention was discussed in collaboration with senior clinical managers in the service, who were consulted regularly throughout the intervention and who provided support. Clinical staff members were encouraged to openly discuss their views and concerns in the training sessions. Ethics approval was granted by the Prince of Wales (POW) Hospital Human Research Ethics Committee (HREC ref no: 18/013 (LNR/18/POWH/13), date of approval: 01/03/2018).

System change interventions were delivered according to the following key practices suggested by the Agency for Healthcare Research and Quality [19]:

*1. Identification of smokers:* Routine screening for tobacco use at the time of admission and documentation in the patient’s electronic health record. In this study, in order that the recording of smoking status could be detected in administrative data, it was necessary to record smoking as a diagnosis (mental and behavioral disorders due to use of tobacco (ICD-10, F17) in the electronic medical record. This marked a change from current practice, where, if recorded, smoking status was typically documented in the body of the clinical notes, and therefore would not register in administrative datasets. This diagnosis could be documented by any member of the treating team (consultant psychiatrist, psychiatry registrar, clinical nurse consultant or staff nurse).

*2. Training:* All staff were encouraged to attend training sessions organized during the intervention phase. Training for staff across all disciplines (medical, nursing and allied health) was provided in nine face-to-face sessions and in presentations at hospital grounds in the three weeks prior to the commencement of the intervention and approximately fortnightly throughout. Training covered rationale for smoking cessation, symptoms of nicotine dependence, and advice on how to record smoking status in electronic medical records. 

*3. Dedicated staff for smoking cessation treatment:* A smoking cessation champion (SCC) was appointed to coordinate staff education and smoking cessation activities and resources (e.g., pharmacotherapy, leaflets, etc.). A clinical nurse consultant (SB) provided dedicated time (0.4 full time equivalent) to act in the role of the SCC. 

*4. Promote hospital policies that support smoking cessation:* Several hospital policies were implemented including the placement of “no smoking” signs, restricting smoking within hospital grounds to designated smoking areas, and developing ward-based “no smoking” policies such as not permitting cigarettes on the unit with provision to lock cigarettes and lighters etc., in personal lockers. 

*5. Tobacco dependence treatment:* Supply of pharmacotherapy and referral to smoking cessation clinics. In this pilot study, there was initial focus only on the provision of nicotine replacement therapy (NRT) by transdermal patches, or in combination with oral gum and inhalers. It was agreed that the prescription of other pharmacotherapy, delivery of brief interventions (motivational interviewing, counselling, etc.) and referral to smoking cessation clinics (while inpatient and/or on discharge) should be delivered in future phases of the project.

### 2.2. Data Collection

Demographic information including age, gender, diagnosis, and country of birth was collated from medical records for all individuals who were inpatients in the three-month period prior to the commencement of the intervention (pre-intervention) and for all individuals who were inpatients in the three-month period following the commencement of the intervention (post-intervention). Information on outcomes was collected from medical records (number of patients with smoking status recorded at any time while inpatient and number of patients prescribed smoking cessation interventions (nicotine replacement therapy, varenicline, buproprion, other) during inpatient stay) and from hospital incident reporting records (number of violent incidents on the unit (aggression to others or to property) as recorded in clinical incident reporting system).

### 2.3. Statistical Analysis

For normally distributed variables, the mean, standard deviation (SD) and range were conducted, otherwise the median and range were presented. Fisher’s exact test was applied for comparisons of categorical variables pre- and post-intervention. Mann–Whitney U test was applied for group comparisons of non-normally distributed data. All analyses were conducted using IBM SPSS 24 (IBM Corp, Armonk, NY, USA).

## 3. Results

During the three-month intervention, there were 214 inpatients, of whom 57.5% were male. The majority had psychosis spectrum illnesses (pre-intervention 67.1%; post-intervention 61.0%, *p* = 0.51). The clinical and demographic characteristics of the inpatient population did not differ significantly in the three months pre- and post-intervention (Table 1). 

At baseline, recording of smoking status in medical records was 1.9%, which increased significantly (*p* ≤ 0.001) to 11.2% post-intervention. NRT was prescribed at baseline to 11.0% of inpatients, which increased significantly (*p* ≤ 0.001) to 26.8% post-intervention (Table 1). The median number of violent incidents recorded per month decreased from 41.0 (SD = 23.3) pre-intervention to 27.0 (SD = 5.0) post-intervention, a difference that was not statistically significant (MWU = 1.00; *p* = 0.13).

## 4. Discussion

The implementation study of smoking cessation on an acute mental health inpatient unit using evidence-based system change interventions achieved significant improvements in routine screening and recording of smoking status in individuals with severe mental illness, and significantly improved prescription of smoking cessation treatment, which more than doubled. There was a non-significant decrease in violent incidents on the unit. These findings indicate that many individuals experiencing severe mental illness are willing to be prescribed smoking cessation treatment and that the inpatient mental health setting provides a key opportunity to offer smoking cessation interventions. While the finding that the number of violent incidents in the post-intervention period was not statistically significantly reduced compared to the pre-intervention period, it may at the very least be interpreted to reassure that this smoking cessation intervention did not increase violence incidents, a commonly-reported fear expressed by mental health staff. The number of violent incidents on the inpatient unit varies widely from month to month, leading there to be a wide range and standard deviation which contributed to the finding not reaching statistical significance. Nonetheless, the finding of reduced violent incidents concords with evidence found in larger studies, where issues such as seasonality could be controlled for [15]. It must be acknowledged, however, that the gains made in this pilot study were modest. Recording of smoking status post-intervention remained very low at 11.2%. While approximately two-thirds of inpatients with severe mental health conditions would be expected to be smokers [11], smoking cessation therapy was prescribed to only 27%. 

In order to achieve equity in physical health for people with mental illness, it is essential that there is routine detection and management of physical health conditions including smoking. Change of culture takes time in complex systems like mental health units, where the issue of smoking remains a divisive issue [20]. Misinformation about smoking is common, but this can be addressed in training which includes evidence that smoking reduces violence on mental health units [15], and that many smokers experiencing severe mental illness both wish to quit and can be supported to do so [10,21]. One limitation of this study is that measures of staff commitment to engaging with the smoking cessation intervention were not systematically measured. This may mean that some staff members who were unable or unwilling to attend did not receive adequate training. Future studies should mandate attendance at training sessions for all staff. A further caveat is that the clinical and demographic information available on the individuals who were inpatient during the intervention was limited. These data may have assisted in providing information about the generalizability of the findings to other inpatient mental health settings internationally. The number of individuals prescribed NRT during the intervention period was considered as a proportion of all inpatients, rather than of all smokers as this information was not known. A final caveat is the brief time frame over which the study was conducted. Acknowledging these limitations, these preliminary data show that effective smoking cessation interventions can be delivered to people with severe mental illness on acute inpatient units with no increase in violence. These findings indicate preliminary success and contribute to a growing and much-needed evidence-base for interventions to address the major health inequities faced by people with severe mental illness [2]. Future implementation phases will prioritize effective participatory collaboration with staff to optimize effectiveness of the intervention, and include additional strategies such as brief intervention training and smoking cessation treatments such as varenicline and buproprion, in addition to NRT, which have been shown to be safe and effective in people experiencing severe mental illness [22]. 

## 5. Conclusions

Smoking and smoking-related harms are common among people with severe mental illness and targeted smoking cessation programmes are much needed in this population. This study highlights that evidence-based smoking cessation interventions can be implemented on an inpatient mental health unit through system change methods. Key initiatives included training of all staff and dedicated smoking cessation staff. Significant improvements were achieved in prescription of smoking cessation treatment to individuals receiving inpatient treatment for severe mental illness. 

## Figures and Tables

**Table 1 medicina-56-00204-t001:** Characteristics of the population, recording of smoking status and smoking cessation treatment prescription pre- and post-intervention.

Characteristic	Pre (n = 209) n (%)	Post (n = 214) n (%)	Fisher’s Exact Test, *p*-Value
Male Gender	108 (51.7)	123 (57.5)	*p* = 0.242
Age			
<35 years	68 (32.5)	86 (40.2)	*p* = 0.575
35–44 years	50 (23.9)	43 (20.1)	
45–54 years	46 (22.0)	41 (19.2)	
55–64 years	31 (14.8)	31 (14.5)	
65+ years	14 (6.7)	13 (6.1)	
Country of Birth			
Australia	151 (72.2)	166 (77.6)	*p* = 0.214
Other	58 (27.8)	48 (22.4)	
Recording of smoking status (n; %)			
Not recorded	205 (98.1)	190 (88.4)	*p* ≤ 0.001
Current smoker	2 (1.0)	18 (8.4)	
Non-smoker	2 (1.0)	6 (2.8)	
Number prescribed nicotine replacement therapy (n; %)	23 (11.0)	56 (26.8)	*p* ≤ 0.001

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
