# Peer review of "Targeted Intervention to Reduce Smoking among People with Severe Mental Illness: Implementation of a Smoking Cessation Intervention in an Inpatient Mental Health Setting"

_medicina, 2020, doi:10.3390/medicina56040204_

Round 1
Reviewer 1 Report
This study deals with a very important topic smoking among mentaly sick who more often than most die from smoking related disorders.
It was a pleasure to read the brief paper that was well written.
I only have a few comments that would improve the paper.
- The first sentence in the Intro says nothing more than what is in the 2nd sentence. Thus please deleted the first.
- Please specify how the outcome variables were measured, i.e. fights, prescription of NRT or actual use.
- Discussion 10th line from bottom is said "lack of a control population". I understood from the Methods that there was an assessment of the outcome variables 3 months before the intervention. Is that not a kind of control?
- Why was not the most efficacious treatment for not smoking Varenicline used?
Author Response
Please find responses in attached Word document.
Reviewer 1
This study deals with a very important topic smoking among mentaly sick who more often than most die from smoking related disorders. It was a pleasure to read the brief paper that was well written. I only have a few comments that would improve the paper. We are very grateful to the reviewer for the above complimentary comments about the paper.
- The first sentence in the Intro says nothing more than what is in the 2nd sentence. Thus please deleted the first. As requested, the first sentence has been removed. (Page 1, Introduction).
- Please specify how the outcome variables were measured, i.e. fights, prescription of NRT or actual use. Thank you for this comment. We have amended the following sentences in Methods to provide additional explanations of how outcome variables were measured, "Information on outcomes was collected from medical records [number of patients with smoking status recorded at any time while inpatient and number of patients prescribed smoking cessation interventions (nicotine replacement therapy, varenicline, buproprion, other) during inpatient stay] and from hospital incident reporting records [number of violent incidents on the unit (aggression to others or to property) as recorded in clinical incident reporting system]." (Page 3, Ln 40).
- Discussion 10th line from bottom is said "lack of a control population". I understood from the Methods that there was an assessment of the outcome variables 3 months before the intervention. Is that not a kind of control? The reviewer is correct that the population who were inpatient in the ward three months before did act as control. We were acknowledging here that this is not a gold standard control, as it did not age or gender match (for example). We have removed the phrase from the Discussion.
- Why was not the most efficacious treatment for not smoking Varenicline used? Both varenicline and buproprion were available for use, but the data reflect that they were not prescribed by clinicians on the inpatient unit over the course of this observational study. For this reason, in the Discussion we make the point that future implementation strategies on the inpatient unit will emphasise the benefits of their use.
Reviewer 2 Report
- Please expand on your hypothesis and literature review on how nicotine smoking cessation and NRT could potentially affect violence on inpatient basis. Any explanation on decrease in number of violent episodes post-intervention but not reaching statistically significant. Is there need to define violent episode?
- Please clarify what hospital staff makes a clinical diagnosis of Tobacco Use Disorder, documents it in the electronic record, documents smoking status in the electronic record and prescribes and monitor NRT (psychiatrist, internist, etc).
- it is not clear if documentation of smoking status increased in the post-trial due to more frequent and diligent documentation in the records by one clinical discipline (psychiatrist, for ex) or due to the fact that more clinical staff members were involved in the documentation( nurses, SW, etc?)
- referral to the Smoking cessation clinic as mentioned for the future phases of the project would be offered to the patients while they are inpatient or after discharge.
- Please clarify "smoking status" and how it was recorded, frequency.
Author Response
Please find responses in attached Word document.
Reviewer 2
- Please expand on your hypothesis and literature review on how nicotine smoking cessation and NRT could potentially affect violence on inpatient basis. Thank you for this comment. A number of sentences have been added to the Introduction, “Concerns that there may be increased aggression, for example because cigarettes may no longer be used to assist in informal de-escalation and behavioural management, or because smokers will quickly experience symptoms of nicotine withdrawal including agitation, have not been upheld. Indeed, there is evidence from a growing number of studies conducted in inpatient mental health settings that both physical and verbal violence decrease following the introduction of smoke-free policies”. (Page 2, Ln 27).
Any explanation on decrease in number of violent episodes post-intervention but not reaching statistically significant. Thank you. To address this point, the following sentences have been added to the Discussion, “While the finding that number of violent incidents in the post-intervention period was not statistically significantly reduced compared to the pre-intervention period, it may at the very least be interpreted to reassure that this smoking cessation intervention did not increase violence incidents, a commonly-reported fear expressed by mental health staff. The number of violent incidents on the inpatient unit varies widely from month to month, leading there to be a wide range and standard deviation which contributed to the finding not reaching statistical significance. Nonetheless, the finding of reduced violent incidents concords with evidence found in larger studies, where issues such as seasonality could be controlled for [15].” (Page 4, Ln 20).
Is there need to define violent episode? Thank you. This has now been provided in more detail in Methods. "Information on outcomes was collected…from hospital incident reporting records [number of violent incidents on the unit (aggression to others or to property) as recorded in clinical incident reporting system]." (Page 3, Ln 40).
Please clarify what hospital staff makes a clinical diagnosis of Tobacco Use Disorder, documents it in the electronic record, documents smoking status in the electronic record and prescribes and monitor NRT (psychiatrist, internist, etc). Thank you for this comment. This diagnosis could be made and documented by any member of the treating mental health staff team (consultant psychiatrist, psychiatry registrar, clinical nurse consultant or staff nurse). This has been acknowledged in the Methods Section with the following sentence, “This diagnosis could be documented by any member of the treating mental health staff team (consultant psychiatrist, psychiatry registrar, clinical nurse consultant or staff nurse).” (Page 3, Ln 14).
- it is not clear if documentation of smoking status increased in the post-trial due to more frequent and diligent documentation in the records by one clinical discipline (psychiatrist, for ex) or due to the fact that more clinical staff members were involved in the documentation( nurses, SW, etc?) Unfortunately, it was not possible to examine this question as the data we had available did not permit the analysis, nor was it one of our hypotheses. Rather, the goal through the system change interventions was to achieve change in the whole staff team (the system).
- Referral to the Smoking cessation clinic as mentioned for the future phases of the project would be offered to the patients while they are inpatient or after discharge. This could be provided while inpatient and/or on discharge. The relevant sentence in Methods has been amended to reflect this, “It was agreed that the prescription of other pharmacotherapy, delivery of brief interventions (motivational interview, counselling, etc.) and referral to smoking cessation clinics (while inpatient and/or on discharge) should be delivered in future phases of the project.” (Page 3, Ln 34).
- Please clarify "smoking status" and how it was recorded, frequency. This refers to the individuals who were identified as smokers, as per Method, point 1: Identification of smokers: Routine screening for tobacco use at the time of admission and documentation in the patient’s electronic health record. In this study, in order that the recording of smoking status could be detected in administrative data, it was necessary to record smoking as a diagnosis (mental and behavioural disorders due to use of tobacco (ICD-10, F17) in the electronic medical record.
Reviewer 3 Report
Thank you for giving me the opportunity to review the article. The authors conducted a study on the targeted intervention to reduce smoking among people with severe mental illness. The topic is clinically and socially important. However, there were fundamental problems of the study methods. Therefore, I thought that the manuscript cannot be accepted for publication as an article (short report). I listed the comments below.
Comments:
Abstract:
- The number of study participants should be added.
- This study is an exploratory research on the targeted intervention to reduce smoking among people with severe mental illness. Therefore, the authors should mention about the requirement of future investigations.
Introduction:
- Clinical studies which related to this study should be mentioned in this section.
Materials and Methods:
- Why did the authors not set any exclusion criteria?
- Why did the authors collect (or use) only limited number of clinical backgrounds?
- The authors set the training session of the intervention, but they only encouraged to attend. It may mean that several staff members did not attend the session, and the process did not control well.
- The details of the intervention should be described.
- The name of software used for statistical analysis should be added.
Results:
- The authors should use the data which obtained both pre- and post-intervention time point.
- Why did the authors compare the demographic characteristics between pre- and post-intervention time point (e.g. age, and country of birth)?
Author Response
Please find responses also in attached Word document.
Reviewer 3
- Abstract: The number of study participants should be added. Thank you, this has been added. (Page 1, Ln 4).
- Abstract: This study is an exploratory research on the targeted intervention to reduce smoking among people with severe mental illness. Therefore, the authors should mention about the requirement of future investigations. Thank you, the following sentence has been added to the Abstract, Conclusions, “Future studies should prioritize effective participatory collaboration with staff to optimize effectiveness of interventions, and should include additional strategies such as brief intervention training and smoking cessation treatments such as varenicline and buproprion in addition to NRT.” (Page 1, Ln 16).
- Introduction: Clinical studies which related to this study should be mentioned in this section. We have included an additional number of sentences citing a number of relevant studies. In the interest of wordcount, this section has been removed form the Discussion in order to now appear, as requested, in the Introduction, “ Thus there is a need to recognize that inpatient settings where smoking is commonplace can often reinforce smoking behaviours and the consequent smoking-related harms, and result in relapse to smoking in persons who have previously quit [18]. Informal ward practice may contribute to this through use of cigarettes and smoking breaks to deescalate acute distress and to reward behaviours such as medication compliance [18]. Staff training has been highlighted as a key factor in the success of smoke-free initiatives [19], while obstacles include both a lack of consistency in applying policies, and a culture of acceptance of smoking behaviours [20].” (Page 2, Ln 32).
- Materials and Methods: Why did the authors not set any exclusion criteria? This was a study conducted at a service (inpatient unit) level and therefore applied to all individuals who were inpatient on the mental health unit at the time of the study. Therefore, no individuals were excluded (exclusion criteria did not apply).
- Materials and Methods: Why did the authors collect (or use) only limited number of clinical backgrounds? These were the data available for the inpatients who were on the mental health unit at the time. These data are those recorded in the electronic medical record.
- Materials and Methods: The authors set the training session of the intervention, but they only encouraged to attend. It may mean that several staff members did not attend the session, and the process did not control well. Thank you for this comment. It is indeed a limitation of the study which we acknowledge in the following sentence which has been added to the Discussion, “This may mean that some staff members who were unable or unwilling to attend did not receive adequate training.” (Page 4, Ln 39).
- Materials and Methods: The details of the intervention should be described. The interventions are extensively described in the Materials and Methods section: System change interventions were delivered according to the following key practices suggested by the Agency for Healthcare Research and Quality [16]:
- Identification of smokers: Routine screening for tobacco use at the time of admission and documentation in the patient’s electronic health record. In this study, in order that the recording of smoking status could be detected in administrative data, it was necessary to record smoking as a diagnosis (mental and behavioral disorders due to use of tobacco (ICD-10, F17) in the electronic medical record. This marked a change from current practice, where, if recorded, smoking status was typically documented in the body of the clinical notes, and therefore would not register in administrative datasets. This diagnosis could be documented by any member of the treating team (consultant psychiatrist, psychiatry registrar, clinical nurse consultant or staff nurse).
- Training: All staff encouraged to attend training sessions organized during intervention phase. Training for staff across all disciplines (medical, nursing and allied health) was provided in nine face-to-face sessions and in presentations at hospital grand rounds in the three weeks prior to the commencement of the intervention and approximately fortnightly throughout. Training covered rationale for smoking cessation, symptoms of nicotine dependence, and advice on how to record smoking status in electronic medical records.
- Dedicated staff for smoking cessation treatment: A Smoking Cessation Champion (SCC) was appointed to coordinate staff education and smoking cessation activities and resources (e.g., pharmacotherapy, leaflets, etc.). A Clinical Nurse Consultant (SB) provided dedicated time (0.4 full time equivalent) to act in the role of the SCC.
- Promote hospital policies that support smoking cessation: Several hospital policies were implemented including the placement of “no smoking” signs, restricting smoking within hospital grounds to designated smoking areas, and developing ward-based “no smoking” policies such as not permitting cigarettes on the unit with provision to lock cigarettes and lighters etc in personal lockers.
- Tobacco dependence treatment: Supply of pharmacotherapy and referral to smoking cessation clinics. In this pilot study, there was initial focus only on the provision of nicotine replacement therapy (NRT) by transdermal patches, or in combination with oral gum and inhalers. It was agreed that the prescription of other pharmacotherapy, delivery of brief interventions (motivational interview, counselling, etc.) and referral to smoking cessation clinics (while inpatient and/or on discharge) should be delivered in future phases of the project.
- Materials and Methods: The name of software used for statistical analysis should be added. Thank you, this sentence has been added, “All analyses were conducted using IBM SPSS 24. (Page 3, Ln 40)
- Results: The authors should use the data which obtained both pre- and post-intervention time point. Results by pre- and post-intervention timepoint are all included and reported in Results and displayed in Table 1.
- Results: Why did the authors compare the demographic characteristics between pre- and post-intervention time point (e.g. age, and country of birth)? We conducted these analyses to demonstrate that there were no systematic differences between the populations of individuals who were inpatient on the mental health unit at the time of the intervention, compared to those who had been inpatient in the 3-month period before the intervention was introduced. This was to demonstrate that there were no demographic differences in the two populations that might be an alternative explanation (confounding factor) for why outcomes improved in the period during which the intervention was in place.
Round 2
Reviewer 3 Report
Thank you for giving me the opportunity to review the revised version of this article. The authors revised the manuscript according to the comments by the reviewers partially. However, there were still fundamental methodological problems. Therefore, I thought that the manuscript cannot be accepted for publication in the journal. I listed the additional comments below.
AR, Authors’ reply; AC, Additional comment
Materials and Methods:
- Why did the authors collect (or use) only limited number of clinical backgrounds?
AR: These were the data available for the inpatients who were on the mental health unit at the time. These data are those recorded in the electronic medical record.
AC: If the authors use the EMR for this study, they should be able to collect the data such as comorbidities, prescribed medications, and other related information of this study. There were no information excluding gender, age, country of birth and the number of prescribed nicotine replacement therapy, the reviewer and potential readers cannot think about other factors which can influence the results.
- The authors set the training session of the intervention, but they only encouraged to attend. It may mean that several staff members did not attend the session, and the process did not control well.
AR: Thank you for this comment. It is indeed a limitation of the study which we acknowledge in the following sentence which has been added to the Discussion, “This may mean that some staff members who were unable or unwilling to attend did not receive adequate training.” (Page 4, Ln 39).
AC: Thank you for adding the information (limitation) about the attendance of the training session. However, there were no information such as the attendance rate or other information which can be evaluate the quality of the process objectively.
Author Response
- Why did the authors collect (or use) only limited number of clinical backgrounds?
AR: These were the data available for the inpatients who were on the mental health unit at the time. These data are those recorded in the electronic medical record.
AC: If the authors use the EMR for this study, they should be able to collect the data such as comorbidities, prescribed medications, and other related information of this study. There were no information excluding gender, age, country of birth and the number of prescribed nicotine replacement therapy, the reviewer and potential readers cannot think about other factors which can influence the results.
We agree with the reviewer that this lack of additional information on demographic and clinical characteristics would have provided helpful descriptive information about he cohort. It was, however, not available routinely in eMR notes. We acknowledge this as a limitation of the study. We are also uncertain that it would significantly influence the results: in this study, we examined whether it is feasible to introduce a smoking cessation programme on an inpatient mental health unit - this research question is addressed by the data presented. The clinical and demographic characteristics of the clients were provided only to provide descriptive information about the clients who were inpatient at the time. They do not affect the analyses.
- The authors set the training session of the intervention, but they only encouraged to attend. It may mean that several staff members did not attend the session, and the process did not control well.
AR: Thank you for this comment. It is indeed a limitation of the study which we acknowledge in the following sentence which has been added to the Discussion, “This may mean that some staff members who were unable or unwilling to attend did not receive adequate training.” (Page 4, Ln 39).
AC: Thank you for adding the information (limitation) about the attendance of the training session. However, there were no information such as the attendance rate or other information which can be evaluate the quality of the process objectively.
The reviewer is correct that these are limitations of the intervention. We have acknowledged this further by adding the following phrase to the Discussion, "Future studies should mandate attendance at training sessions for all staff".